# The Pharmacokinetic Profile and Bioavailability of Enteral N-Acetylcysteine in Intensive Care Unit

**DOI:** 10.3390/medicina57111218

**Published:** 2021-11-08

**Authors:** Kersti Teder, Liivi Maddison, Hiie Soeorg, Andres Meos, Juri Karjagin

**Affiliations:** 1Institute of Pharmacy, University of Tartu, Nooruse 1, 50411 Tartu, Estonia; andres.meos@ut.ee; 2Pharmacy Department, Tartu University Hospital, L. Puusepa 8, 50406 Tartu, Estonia; 3Institute of Clinical Medicine, University of Tartu, L. Puusepa 8, 50406 Tartu, Estonia; liivi.maddison@kliinikum.ee (L.M.); juri.karjagin@kliinikum.ee (J.K.); 4Clinic of Anaesthesiology and Intensive Care, Tartu University Hospital, L. Puusepa 8, 50406 Tartu, Estonia; 5Department of Microbiology, Institute of Biomedicine and Translational Medicine, University of Tartu, Ravila 19, 50411 Tartu, Estonia; hiie.soeorg@ut.ee

**Keywords:** N-acetylcysteine, pharmacokinetics, bioavailability, infection prevention, pneumonia

## Abstract

*Background and Objectives*: N-acetylcysteine (NAC) is a mucolytic agent used to prevent ventilator-associated pneumonia in intensive care units. This study aimed to evaluate the oral bioavailability of NAC in critically ill patients with pneumonia, isolated acute brain injury and abdominal sepsis. *Materials and Methods*: This quantitative and descriptive study compared NAC’s pharmacokinetics after intravenous and enteral administration. 600 mg of NAC was administered in both ways, and the blood levels for NAC were measured. *Results*: 18 patients with pneumonia, 19 patients with brain injury and 17 patients with abdominal sepsis were included in the population pharmacokinetic modelling. A three-compartmental model without lag-time provided the best fit to the data. Oral bioavailability was estimated as 11.6% (95% confidence interval 6.3–16.9%), similar to bioavailability in healthy volunteers and patients with chronic pulmonary diseases. *Conclusions*: The bioavailability of enteral NAC of ICU patients with different diseases is similar to the published data on healthy volunteers.

## 1. Introduction

N-acetylcysteine (NAC), a precursor of cysteine and glutathione, is a beneficial mucolytic agent [1,2]. Since its emergence in the 1960s, it has been studied and used to treat chronic bronchitis and other chronic pulmonary diseases (CPD) complicated by viscous mucus production [1,2,3,4,5]. Recent studies have underlined antioxidant properties of NAC as an additional benefit in treating pneumonia in elderly age groups [6,7,8]. NAC is also used to prevent ventilator-associated pneumonia (VAP), with the effectiveness of preventing VAP in critically ill populations most likely depending on the administration route [9,10]. Furthermore, NAC has been used as an antidote to paracetamol poisoning and has benefited many other conditions [2,11,12,13,14,15,16,17].

For the mucolytic effect, the first NAC formulations were inhalations [18]. Not long after, NAC was also proven to be equally effective in peroral (PO) or parenteral administration [3,19]. Compared to oral administration, the mucolytic effect of intravenous (IV) NAC has been proven to have a more rapid onset of action, but there is no significant difference in long term effects [3]. However, this might suggest that in acute situations, such as preventing VAP or treating patients with established pneumonia, NAC’s administration should be IV.

Some studies have been conducted on NAC pharmacokinetics (PK) in healthy volunteers and patients with CPD [13,18,20,21,22,23]. Absolute bioavailability of oral NAC has been proven to be low, varying between 6–10%, probably due to extensive first-pass metabolism in the gut wall and liver [20,21,22]. The studies on NAC’s clinical efficacy in critically ill patients showed positive VAP incidence and secretions [2,9,10], but these studies lack PK data.

Critically ill patients in intensive care units (ICUs) have different pathologies that might change the PK of the medicines, including absorption of enteral medicines [24,25,26,27,28,29,30]. It appears that mean serum NAC levels are higher in patients with chronic liver disease [31] and end-stage renal disease [32]. Patients with kidney injury have shown that although hemofiltration does not affect PK of NAC, haemodialysis does [33]. An additional issue is the extensive use of medicines in the ICU; the significance of drug-drug interactions with NAC is uncertain [22,34,35,36]. Aspects of absorption of enteral medicines administered through feeding tubes are also not well studied. Therefore, considering the problems with critically ill patients’ gastrointestinal tract (GIT), one can assume that absorption of medicine might change [29,30].

The study aimed to evaluate and describe NAC’s oral bioavailability in ICU patients by including three patient groups: patients with established pneumonia, patients with isolated acute brain injury and patients with abdominal sepsis.

## 2. Materials and Methods

This is a prospective, quantitative and descriptive study that describes PK of NAC after IV and enteral administration. The study protocol was approved by the Ethics Review Committee on Human Research of the University of Tartu (protocol number 204/T-8, approval date: 16 May 2011) and was registered in the EudraGMP database (EudraCT no 2011-002093-23, approval date: 21 September 2011). The study was performed according to the Declaration of Helsinki and following ICH Good Clinical Practice. Before inclusion, informed consent from the patient or next of kin was obtained. Patients were enrolled in the study from September 2012 to January 2017.

### 2.1. Patients

Adult patients admitted to the Department of 2nd Intensive Care Unit at Tartu University Hospital were screened for the study inclusion criteria. The patients with established pneumonia (the lung group) or with isolated acute brain injury of traumatic and non-traumatic origin (the brain group) or with abdominal sepsis (the gut group) and NAC indicated for mucolytic effect for preventing VAP or treating pneumonia were included in the study. Exclusion criteria were the patient’s or next of kin’s refusal to participate, allergy to NAC or excipients of the medicine, the patients with intolerance or contraindication to enteral nutrition or medicines.

### 2.2. NAC Formulations

The drug formulations used in the study were those used during the treatment in the hospital. NAC injection solution (Sandoz Pharmaceuticals ACC; 100 mg/mL, 3 mL) [37] was used for IV administration, and NAC powder for oral solution (Sandoz Pharmaceuticals ACC; 200 mg; three bags (at each time)) [38] was used for enteral administration.

### 2.3. Study Procedures

After obtaining informed consent, the patient was included in the study. If the patient had been given NAC before, a wash-out period of at least 24 h was implemented. A decision of implementing the wash-out period of twenty-four hours was based on the previous studies, which showed that the elimination half-life (T_1/2_) of NAC is about six hours (5.6 h after IV administration and about 6.3 h after enteral administration) [21,22].

On the first day of the study, all the patients were administered a single bolus dose of IV NAC (600 mg diluted in 10 mL saline), and blood samples for PK analysis were taken ten times (before the drug administration and at time points following drug administration: 10 min, 20 min, 40 min, 1 h, 1.5 h, 2 h, 3 h, 4 h and 5 h). The following day, 24 h after the first dose, enteral NAC (600 mg dissolved in water) was administered through the nasogastric tube or orally if the patient was capable. Again, blood samples for the analysis were taken 15 times (before the drug administration and at time points following drug administration: 20 min, 40 min, 1 h, 1.5 h, 2 h, 2.5 h, 3 h, 3.5 h, 4 h, 4.5 h, 5 h, 6 h, 7 h and 8 h).

The bedside nurse collected blood samples from the arterial cannula, a part of the critically ill patients’ routine monitoring in the ICU.

### 2.4. Analysis of the Blood Samples

The blood samples were collected in Lithium Heparin tubes, immediately centrifuged with Hettich EBA 20 (Andreas Hettich GmbH & Co. KG, Tuttlingen, Germany) centrifuge (4000 rpm; 10 min), and the plasma was transferred into Eppendorf’s tubes. The plasma tubes were kept at −80 °C for further high-pressure liquid chromatography (HPLC) analyses. In addition, each plasma sample was stored in three sets to ensure a possibility to conduct repetitive HPLC analysis if needed.

NAC was determined from plasma samples by the HPLC method after NAC derivatisation at the University of Tartu Institute of Pharmacy. The NAC derivatisation and HPLC analysis methods were performed as previously described [39,40,41], and only quantities of the solutions were changed.

For the NAC derivatisation, 200 µL of plasma was mixed with 50 µL of 0.8 M phosphate buffer, 20 µL of 0.2 mM 3,3′-dithiodipropionicacid solution and 20 µL of 0.25 M Tris(2-carboxyethyl)phosphine solution in phosphate buffer and incubated for 10 min. After ten minutes, 19 µL of 0.1 M 2-chloro-1-methylquinolinium tetrafluoroborate solution in water was added to the above solution and was mixed well. After two minutes of incubation, 50 µL of 8.5 M perchloric acid was added, and the solution was remixed. Finally, the solution was centrifuged for 15 min at 13,000 rpm, and 200 µL of clear supernatant was transferred into HPLC tubes for analysis.

HPLC analysis was conducted with Shimadzu LC20 chromatography (Shimadzu Corporation, Kyoto, Japan). For this, 50 µL of supernatant was injected into the Luna2 (C18, 150 mm × 4.6 mm, 5 µm) column (Phenomenex, Torrence, California, USA). Mobile phase (binary high-pressure gradient, flow rate 1.2 mL/min; temperature 25 °C) was a combination of two eluents, 0.07 M trichloroacetic acid buffer (solution A; pH adjusted to 1.65 with the solution of lithium hydroxide) and acetonitrile (solution B). The gradient of these solutions was as follows: 0–4 min 11% solution B, 4–8 min 11–30% solution B, 8–12 min 30–11% solution B and 12–15 min 11% solution B. NAC was detected with diode-array detector SPD-M20A (Shimadzu Corporation, Kyoto, Japan) at the wavelength of 355 nm. A quantification limit was 0.13 mg/L. The HPLC analysis process was controlled by software LabSolutions (version 5.71 SP1, Shimadzu Corporation, Kyoto, Japan).

### 2.5. Statistical Analysis

All the patients’ data and results from HPLC analysis were transferred to Microsoft Excel for Microsoft 365 (version 2110; Microsoft Corporation, Los Angeles, CA, USA). Statistical analysis was performed in R software (version 4.0.5; The R Foundation, Vienna, Austria). For comparing continuous variables between the study groups, the Kruskal-Wallis test was used, followed by Dunn’s test for multiple pairwise comparisons [42].

Population pharmacokinetic modelling was carried out by nonlinear mixed-effects modelling in NONMEM (version 7.43; ICON Development Solutions, MD, Gaithersburg, MD, USA). IV and oral NAC administration data were analysed simultaneously. For the patients with unknown dosing history, the central compartment was initialised by pre-dose sample concentration (i.e., compartment initialisation method) [43]. Concentrations below the lower limit of quantification (0.13 mg/L) (LLOQ) were included in the model as half of the LLOQ. One, two- and three-compartment structural models with and without lag-time were tested. Interindividual variability was tested for all parameters assuming log-normal distributions, except for oral bioavailability.

Each study group was tested as a covariate and retained in the model if it decreased an objective function value by at least 3.84 (statistical significance level 0.05). Oral bioavailability was included in the model by logit transformation (1/(1 + EXP(−(F + IIV_F_))), where F is a parameter for bioavailability and IIV_F_ its interindividual variability) to ensure its value between 0 and 1 [44]. Standard error for F was calculated by the delta method. For residual variability, proportional, additive and combined additive and proportional error models common to and separately for oral and IV data were tested. The fit of the model was assessed by goodness-of-fit plots (observations vs population or individual predictions, conditional weighted residuals vs time or predictions, individual weighted residuals vs time or predictions) and prediction-corrected visual predictive check stratified by route of administration.

## 3. Results

### 3.1. Characteristics of the Study Population

Altogether 59 patients were included in the study: 18 patients in the lung group, 22 patients in the brain group and 19 patients in the gut group. Five patients were excluded from the final analysis; among the five patients, two patients died, and one had a gastrectomy before enteral administration was carried out. The remaining two patients had remarkably different NAC blood level changes than the rest of the patients. The latter two patients had no apparent clinical conditions that would have explained the difference, and a pre-analytical error was suspected. Thus, 54 patients were included in the final analysis. The patients’ data were tested with the Kruskal-Wallis test and showed no statistically significant differences between the three groups; the data is presented in Table 1.

An overview of the patients’ body fluid balance and lab tests are presented in Table 2. Dunn’s test showed some statistically significant differences among the study groups. For example, the gut group patients’ IV and PO liquids differed significantly from the other groups; the gut group patients had more IV and lower PO liquids. This was expected since the gut group’s gastrointestinal failure score (GIF-score) [45,46] values were also significantly higher than the other study groups, referring to malfunctioning GIT. In addition, albumin levels of the brain group patients were higher, and urea levels were lower (statistically significantly on the second day) than the other group patients. Other statistically significant differences, mainly between brain and gut groups, are presented in Table 2.

### 3.2. Characteristics of the Lung Group

The lung group consisted of 18 patients with pneumonia. In ten patients, pneumonia was the primary reason for ICU hospitalisation. In eight patients, pneumonia was developed as a complication during the hospital stay. Co-morbidities of the patients included neurological disease in seven patients and intoxication in one patient with secondary pneumonia.

IV noradrenaline was used in 16 patients to maintain sufficient mean arterial pressure; specifically, in 14 patients, the drug was used on both days of the study, and in two patients, it was used only on the first day. Noradrenaline dose range was between 0.01–0.34 µg/kg/min on the first day and 0.01–0.61 µg/kg/min on the second day of the study. In addition, antibiotics were administered in all (18) patients by IV route, and one patient additionally had enteral vancomycin. Seventeen patients had their medicines and fluids administered both IV and enterally (by mouth or through an NGFT); they were also fed enterally.

According to GIF-score, eight patients had normal GIT function (GIF-score was 0 on both days), and seven patients had impaired GIT function (GIF-score of one on one (four patients) or both (three patients) of the study days). In addition, three patients had GIF-score above two at least on one day of the study, and one of these patients had all medicines, fluids, and feed by IV.

### 3.3. Characteristics of the Brain Group

The brain group consisted of 19 patients. Most patients of the brain group had a traumatic brain injury with subdural haemorrhage in 11 patients, epidural haemorrhage in one patient and combined injuries in six patients. In addition, one patient had an ischaemic stroke. Out of 19 patients, 14 patients underwent injury-related surgery; 12 patients had a craniotomy, one patient had ventriculostomy, and another patient underwent parenchymal ICP sensor placement.

To maintain sufficient brain perfusion pressure, IV noradrenaline was used in 13 patients; in specific terms, seven patients were given the drug on both days; for five patients, it was used on the first day, and for one patient, it was given only on the second day of the study. Noradrenaline dose range was between 0.01–0.36 µg/kg/min on the first day and 0.05–0.34 µg/kg/min. In addition, antibiotics were administered IV in 17 patients (one had antibiotics only on the first day of the study). Drugs and fluids were administered both IV and enterally through an NGFT. Eighteen patients had one or more enteral medications on the second day of the study when NAC was administered through a feeding tube or orally.

All 19 patients were fed enterally (by mouth or by feeding tube), and according to GIF-score, most of the patients’ GIT had a normal function (ten patients had GIF-score of 0 on both days, six patients had GIF-score of one on the first day and 0 on the second day, and three patients had GIF-score of one on both days).

### 3.4. Characteristics of the Gut Group

The gut group consisted of 17 patients with abdominal sepsis of different origins. In specific terms, there were four patients with postoperative intestinal obstructions, ten patients of peritonitis from perforations of GIT (two patients were with perforation in upper GIT and eight patients were with perforations in the lower GIT), three patients were with acute pancreatitis, one patient was with acute cholangitis, and the remaining one was with *Clostridium difficile* enterocolitis.

To maintain sufficient mean arterial pressure, IV noradrenaline was used in 13 patients; out of these 13 patients, nine patients were administered the drug on both days; in three patients, it was used only on the first day, and in one patient, it was given only on the second day of the study. Noradrenaline dose range was between 0.04–0.43 µg/kg/min on the first day and 0.03–0.56 µg/kg/min. Antibiotics were administered in all 19 patients by IV route; one patient had an additional enteral antibiotic (vancomycin).

Drugs and fluids were administered both IV and enterally through NGFT. Eight patients had one or more enteral medications on the second day of the study when NAC was administered through a feeding tube or orally. Seven patients were fed enterally (by mouth or by NGFT) on both days of the study, and two patients were fed on the second day (when NAC was administered enterally). According to GIF-score, most of the patients’ GIT did not have a normal function; only one patient had a GIF-score of 0 on both days. Most patients (nine) had GIF-score one on both days.

### 3.5. Pharmacokinetics and Bioavailability Evaluation

After IV administration, a median of maximum blood concentrations (C_max_) was 36.1 mg/L (interquartile range: 22.8–46.8 mg/L) on the first measurement of time point (10 min after administration). The median of C_max_ after enteral administration was 2.5 mg/L (interquartile range: 1.5–4.0 mg/L). Time for C_max_ (t_max_) was measured 90 min (interquartile range: 60–120 min) after administration.

Although there were no statistically significant differences among the study groups in the case of C_max_ and T_max_ values after IV and enteral administration, the brain group area under the curve (AUC; calculated using the trapezoidal method) values of five hours after IV administration and eight hours after enteral administration were significantly lower than the other study groups. The result of the different study groups is presented in Table 3.

A three-compartmental model without lag-time provided the best fit to the data. Interindividual variability was retained in all parameters. Estimation of PK model parameters are shown in Table 4. Patients with brain injury had a larger central and peripheral volume of distribution, clearance, intercompartmental clearance and absorption rate as shown by Θ_1_ to Θ_5_ (covariates for the parameters indicating whether the patient had a brain injury or not) values greater than 1.

Final residual error model included proportional and additive error terms for oral and IV data. Visual predictive check (Figure 1) and goodness-of-fit plots (Figure 2 and Figure 3) showed a good fit for the data model. Oral bioavailability was estimated as 11.6% (95% confidence interval 6.3–16.9%), and there was no significant difference among the study groups.

## 4. Discussion

Although there have been numerous studies on NAC PK in healthy volunteers and patients with CPD [13,18,20,21,22], to the best of the authors’ knowledge, this is the first study on NAC PK involving ICU patients with established pneumonia, isolated acute brain injury or abdominal sepsis. The previous studies in critically ill patients were primarily focused on the clinical efficacy of NAC [2,9,10,11,47], and these studies lack PK data. Data about NAC PK in patients with end-stage renal disease showed a significant increase in NAC’s blood concentration levels [32], but these patients were excluded from the present study.

The absolute bioavailability of enteral NAC in patients with established pneumonia, isolated acute brain injury and abdominal sepsis was found to be relatively low, 11.6% (6.3–16.9%) and, therefore, it was similar to healthy volunteers and patients with CPD (varying between 6–10%) [20,21,22]. In addition, median of NAC’s C_max_ after enteral administration was 2.5 mg/L, similar to the previous results (2.3–2.9 mg/L) [13,20,22,23]. The blood concentration-time curve was relatively flat, and the t_max_ was 90 min, which was a bit more than the previous studies on healthy volunteers (40–60 min) [13,20,23].

In the present study, NAC blood levels after IV administration were 36.1 mg/L, approximately 14 times higher than in the case of enteral administration (2.5 mg/L). There was no difference in long term mucolytic effects if NAC was administered orally or IV, but the onset of action and the effect of preventing VAP might be increased if NAC is administered IV [3,9]. In addition, as shown in the previous studies, starting the treatment with NAC by IV administration might decrease the duration of ICU stay because of VAP [3,9]. The suggestion of starting VAP prevention with NAC by IV administration is based on differences in the bioavailability and blood concentrations, but not on a randomised comparison of both routes and therefore needs further investigation.

### 4.1. Group Differences in PK

Although it was not our primary goal to compare the patient groups, there were some interesting observations. The brain group had faster absorption and faster clearance (IV and enteral) of NAC, which resulted in lower exposure. In addition, this study group also had a larger central and peripheral volume of distribution. Of note, all this did not influence bioavailability. The brain group had preserved renal and GI functions and had less inflammation, which could be a combined reason for PK findings, but needs further investigation.

The clearance of total NAC in the brain group was somewhat similar to the previous studies (7.1 L/h). According to the manufacturers’ information and Olsson et al., the clearance of total NAC is 0.11 L/h/kg, or 7.7 L/h for a person weighing 70 kg and the total NAC concentrations decline in a triphasic manner [21,37,38]. However, the lung and gut group clearance was two times lower (3.57 L/h). It was surprising since Borgström et al. (1986) had shown total body clearance of NAC for these groups as 14.5 L/h, while Papi et al. (2021) reported it as 56.5 L/h [20,23]. The reasons for these differences are worthy of further investigation.

### 4.2. Study Limitations

The study had several limitations. First, the patient population was relatively small; on average, there were 18 patients per group due to clinical trial restrictions. However, the total number of patients included in the study was sufficient to estimate oral bioavailability of NAC. Secondly, the wash-out period between the two administrations was 24 h, and this could have been longer since the first IV dose might have influenced the concentration of NAC after enteral administration. However, the influence would have been to increase absolute bioavailability values, which only increases the confidence that NAC’s oral bioavailability is low and similar to the healthy population. Moreover, oral bioavailability was estimated by a PK model that can ensure the remaining concentrations after IV administration. The authors did not consider a more extended wash-out period because the physiology of the critically ill patient changes vastly, and extended periods could influence PK accordingly.

## 5. Conclusions

According to the study findings, our primary outcome measure on the bioavailability of enteral NAC of ICU patients with different diseases was found to be similar to the published data on healthy volunteers. However, the PK analysis showed great variability in concentrations and peak times. Some differences between parameters cannot be clearly explained using routinely registered health parameters; therefore, further studies with more sophisticated designs are needed.

## Figures and Tables

**Figure 1 medicina-57-01218-f001:**
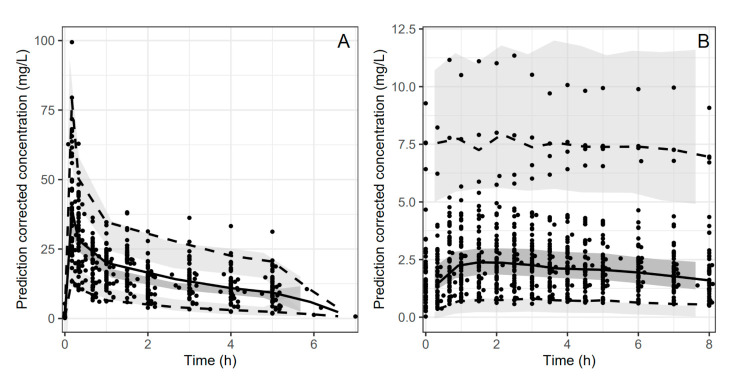
Visual predictive check for peroral (**A**) and intravenous (**B**) data. Open dots show the observations; solid line shows the median, and dashed lines show the 2.5th and 97.5th percentiles, shaded areas the 95% confidence intervals of the corresponding predicted concentrations.

**Figure 2 medicina-57-01218-f002:**
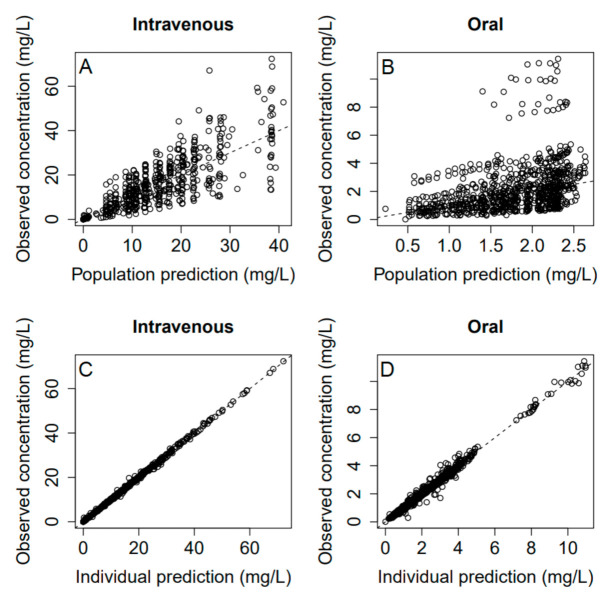
Population (**A**,**B**) and individual (**C**,**D**) predictions versus observations for intravenous (**A**,**C**) and oral (**B**,**D**) data.

**Figure 3 medicina-57-01218-f003:**
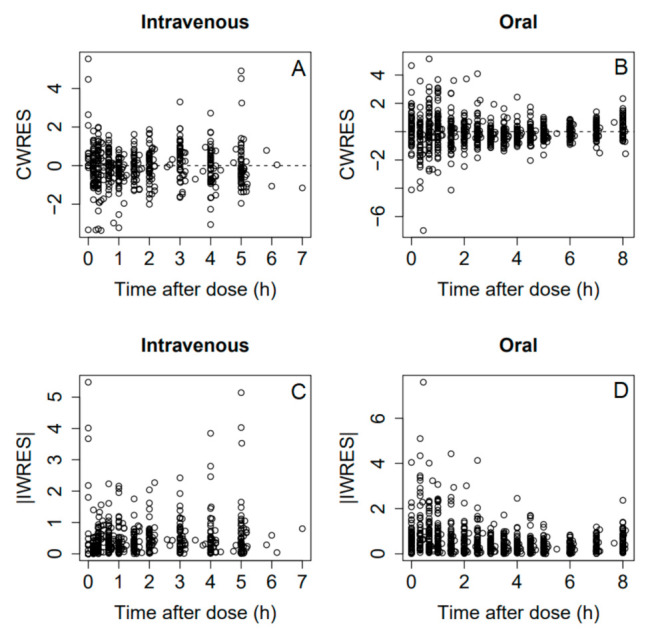
Conditional weighted residuals versus time (**A**,**B**) and absolute value of individual weighted residuals versus time (**C**,**D**) for intravenous (**A**,**C**) and oral (**B**,**D**) data.

**Table 1 medicina-57-01218-t001:** Characteristics of the study patients enrolled in final PK analysis.

Patient Characteristics	Median (Interquartile Range)		
	Lung Group	Brain Group	Gut Group
Number of patients	18	19	17
Sex (male)	13	15	11
Age (years)	65 (59–71)	61 (50–69)	71 (66–78)
Height (cm)	175 (167–180)	175 (170–180)	169 (167–175)
Weight (kg)	78 (71–96)	80 (73–85)	76 (66–90)
BMI (kg/m^2^)	25 (23–29)	26 (24–28)	26 (24–29)
APACHE II score on admission day	22 (19–26)	22 (20–25)	19 (12–23)

PK—pharmacokinetic; BMI—body mass index; APACHE—acute physiology and chronic health evaluation.

**Table 2 medicina-57-01218-t002:** The study patients’ fluid balance and lab test results on the study days.

	Median (Interquartile Range)					
	Lung Group		Brain Group		Gut Group	
	First Day	Second Day	First Day	Second Day	First Day	Second Day
FLUID BALANCE per 24 h						
IV infusion (mL) ^A^	1641 (1116–2941)	1464 (689–2603)	1749 (1415–2935)	1214 (787–2174)	3023 (2248–3495)	2599 (1680–3470)
PO liquids (mL) ^B^	1013 (348–1341)	1061 (731–1286)	840 (425–1195)	890 (390–1275)	0 (0–400)	60 (0–450)
Urine (mL)	2075 (1734–2893)	1863 (1553–2123)	2445 (2105–2860)	1550 (1100–2050)	1800 (1095–2850)	1850 (1570–2200)
NGFT liquids (mL)	25 (0–218)	100 (0–296)	40 (0–263)	0 (0–25)	50 (10–350)	0 (0–750)
Fluid balance (mL)	1044 (93–1452)	388 (119–1350)	630 ((–609)–1533)	550 ((–40)–924)	1188 (337–2285)	1183 (275–1982)
LAB TESTS						
HGB (g/L)	106 (93–124)	102 (92–125)	106 (98–117)	106 (97–118)	103 (92–111)	92 (88–97)
HCT (%)	32 (27–41)	32 (28–39)	31 (30–34)	31 (29–36)	31 (27–33)	28 (27–29)
CRP (mg/L) ^C^	150 (76–168)	132 (91–231)	90 (51–125)	125 (86–179)	185 (157–321)	240 (131–313)
Creatinine (µmol/mL)	81 (44–116)	76 (43–97)	63 (51–97)	53 (49–78)	71 (43–132)	75 (49–124)
eGFR (mL/min/1.73 m^2^) ^D^	87 (60–116)	91 (82–120)	103 (67–122)	108 (86–136)	81 (45–110)	77 (56–104)
Urea (mmol/L) ^E^	9 (4–11)	9 (4–10)	5 (3–7)	5 (3–7)	9 (6–12)	8 (6–11)
Bilirubin (µmol/L)	12 (7–24)	12 (7–26)	8 (6–16)	8 (6–16)	11 (9–20)	12 (10–20)
AST (U/L)	45 (33–74)	45 (33–67)	33 (22–85)	32 (22–65)	30 (19–58)	23 (19–58)
ALT (U/L)	34 (19–66)	36 (19–64)	31 (13–50)	37 (13–52)	15 (11–45)	14 (10–26)
Protein (U/L) ^F^	61 (59–67)	62 (61–64)	61 (58–63)	61 (58–63)	55 (52–60)	55 (54–60)
Albumin (g/L) ^G^	27 (25–30)	29 (27–32)	34 (32–35)	34 (32–35)	26 (25–29)	27 (25–32)
Lactate (mmol/L) ^H^	1.5 (1.1–1.8)	1.4 (1.2–1.8)	1.1 (0.8–1.5)	1.1 (0.8–1.2)	1.3 (1.1–1.6)	1.1 (1.0–1.9)
SOFA score	9 (7–11)	8 (6–12)	8 (6–10)	7 (4–8)	7 (5–11)	6 (4–11)
GIF-score ^I^	1 (0–1)	0 (0–1)	0 (0–1)	0 (0–0)	1 (1–1)	1 (1–2)

HGB—haemoglobin; HCT—haematocrit; CRP—serum concentration of C-reactive protein; eGFR—estimated glomerular filtration rate; AST—aspartate aminotransferase; ALT—alanine amino transaminase; SOFA—sequential organ failure assessment; GIF—gastrointestinal failure. ^A^ Both day values of the gut group differ statistically significant from the other groups (first day: gut-brain *p* = 0.0037; gut-lung *p* = 0.0210; second day gut-brain *p* = 0.0056; gut-lung *p* = 0.0073). ^B^ Both day values of the gut group differ statistically relevantly from the other groups (first day: gut-brain *p* = 0.0027; gut-lung *p* = 0.0003; second day gut-brain *p* = 0.0008; gut-lung *p* = 0.0009). ^C^ Second day values of the brain and the gut group differ statistically relevantly from each other (*p* = 0.0054). ^D^ First day values of the brain and the gut group differ statistically relevantly from each other (*p* = 0.0320). ^E^ Second day values of the brain group differ statistically relevantly from other groups (brain-gut *p* = 0.0290; brain-lung *p* = 0.0361). ^F^ Second day values the gut group differ statistically relevantly from the other groups (gut-brain *p* = 0.0167; gut-lung *p* = 0.0204). ^G^ Both day values of the brain group differ statistically relevantly from other groups (first day: brain-gut *p* = 0.0055; brain-lung *p* = 0.0368; second day brain-gut *p* = 0.0003; brain-lung *p* = 0.0026). ^H^ First day values of the brain and the lung group differ statistically relevantly from each other (*p* = 0.02940). ^I^ Both day values of the gut group differ statistically relevantly from the other groups (first day: gut-brain *p* = 0.00004; gut-lung *p* = 0.0084; second day gut-brain *p* = 0.0037; gut-lung *p* = 0.0111).

**Table 3 medicina-57-01218-t003:** Pharmacokinetic parameters of total NAC after administration of 600 mg NAC.

PK Characteristic	Median (Interquartile Range)		
	Lung Group	Brain Group	Gut Group
C_max_ IV (mg/L)	37.7 (28.4–50.4)	23.6 (17.2–42.0)	40.0 (33.9–53.9)
C_max_ PO (mg/L)	2.7 (1.5–4.3)	1.9 (1.5–2.8)	2.8 (2.3–4.3)
t_max_ PO (min)	120 (50–150)	90 (60–120)	60 (60–120)
AUC_0–5_ IV (mg/L×h)	79 (64–97)	53 (32–59)	98 (54–110)
AUC_0–8_ PO (mg/L×h)	18 (10–24)	9 (6–14)	17 (12–26)

NAC—N-acetylcysteine; PK—pharmacokinetic; C_max_—maximum blood concentrations; IV—intravenous; PO—oral; t_max_—time for C_max_; AUC_0–5_—area under the curve (0–5 h); AUC_0–8_—area under the curve (0–8 h).

**Table 4 medicina-57-01218-t004:** Pharmacokinetic parameters estimated by the final population pharmacokinetic model.

Parameter	Estimate (Standard Error)	IIV ^a^ (Standard Error)	Shrinkage (%)
V (L) = V_0_·Θ_1_ ^brain group^			
V_0_	11.80 (1.44)	56.3 (5.5)	1.8
Θ_1_	1.59 (0.29)	-	-
CL (L/h) = CL_0_·Θ_2_ ^brain group^			
CL_0_	3.57 (0.20)	56.8 (11.5)	0.1
Θ_2_	1.99 (0.28)	-	-
KA (1/h) = KA_0_·Θ_3_ ^brain group^			
KA_0_	0.33 (0.09)	110 (14.5)	6.0
Θ_3_	2.63 (0.98)	-	-
F ^b^	−2.03 (0.07)	46.1 (7.0)	10.1
V2 (L) = V2_0_·Θ_4_ ^brain group^			
V2_0_	12.0 (2.0)	66.9 (21.1)	5.3
Θ_4_	2.07 (0.47)	-	-
Q2 (L/h) = Q2_0_·Θ_5_ ^brain group^			
Q2_0_	4.60 (0.73)	63.2 (13.1)	18.4
Θ_5_	2.16 (0.49)	-	-
V3 (L)	8.62 (0.98)	52.1 (7.7)	10.9
Q3 (L/h)	43.8 (6.46)	55.2 (10.3)	25.5
Residual variability			
Intravenous, proportional (%)	3.4 (1.3)		21.6
Oral, proportional (%)	8.2 (1.8)		9.6
Intravenous, additive (mg/L)	0.3 (0.1)		21.6
Oral, additive (mg/L)	0.07 (0.03)		9.6

CL—clearance; KA—absorption rate; Θ_1_–Θ_5_—brain group differences; Q2, Q3—intercompartmental clearances between the central and first and the second peripheral compartment, respectively; V, V2, V3—volumes of the central, the first and the second peripheral compartment, respectively. ^a^ IIV (interindividual variability) is expressed in terms of coefficient of variation (calculated as SQRT(OMEGA) × 100), except for F, for which standard deviation is presented. ^b^ F is the parameter for bioavailability; bioavailability was calculated as 1/(1 + EXP(−(F + IIV_F_))).

## Data Availability

The datasets generated and analysed during the current study are available with the corresponding author on a reasonable request.

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
