# Peer review of "The Pharmacokinetic Profile and Bioavailability of Enteral N-Acetylcysteine in Intensive Care Unit"

_medicina, 2021, doi:10.3390/medicina57111218_

Round 1
Reviewer 1 Report
In section 1, paragraph 2, line 2, the author mentioned oral (PO). What is PO, Abbreviations in the text must be clear.
In section 1, page 2, line 5, please review the statement again.
In section 2.2, line 7, NAC formulation,the author wrote The drug formulation……. Kindly specify the drug formulation the authors used.
In section 2.4, line 2, the author mentioned EBA. Kindly elaborate the term EBA.
In section3.2, last paragraph, the author mentioned seven patients had some limitations. Kindly specify limitations.
Author mentioned the dose range of noradrenaline of each case study. Kindly mention the reference from where the range selected
In section 5, conclusions are not elaborative. Kindly elaborate further.
In reference 2 and 7, kindly add page number and volume number.
Please again go through referncer 3,39
Author Response
The authors would like to thank the reviewer for the comments and suggestions. The manuscript has been revised and the changes made are described in the file (21.11.01 - answers to reviewers - RW1.docx). All the changes are also visible in the manuscript.

Reviewer 2 Report
The authors explored the topic and they obtained the purpose of the study. This research aimed to evaluate the oral bio-availability of N-acetylcysteine (NAC) in critically ill patients with pneumonia, isolated acute brain injury and ab-dominal sepsis.
- The paper is well written and text is clear to read.
- The methods used are sufficiently documented. Results obtained are well explained and data interpretation is also correct. Conclusions are consistent with the evidence and arguments presented.
- About limitations the authors should completely review references following the guidelines references.
- Journal Articles:
1. Author 1, A.B.; Author 2, C.D. Title of the article. Abbreviated Journal Name Year, Volume, page range. - Books and Book Chapters:
2. Author 1, A.; Author 2, B. Book Title, 3rd ed.; Publisher: Publisher Location, Country, Year; pp. 154–196.
3. Author 1, A.; Author 2, B. Title of the chapter. In Book Title, 2nd ed.; Editor 1, A., Editor 2, B., Eds.; Publisher: Publisher Location, Country, Year; Volume 3, pp. 154–196. - Regarding limitations, the authors themselves describe a too low number of patients and a wash-out period between two administrations too short.
Author Response
The authors would like to thank the reviewer for the comments and suggestions. The manuscript has been revised and the changes made are described in the file (21.11.01 - answers to reviewers - RW2.docx). All the changes are also visible in the manuscript.
